

# Sea urchins: an update on their pharmacological properties

Dulce María Moreno-García[1,*], Monica Salas-Rojas[2],
Eduardo Fernández-Martínez[3], Ma del Rocío López-Cuellar[4],
Carolina G. Sosa-Gutierrez[1], Armando Peláez-Acero[1],
Nallely Rivero-Perez[1], Adrian Zaragoza-Bastida[1] and
Deyanira Ojeda-Ramírez[1,*]

[1] Área Académica de Medicina Veterinaria y Zootecnia. Instituto de Ciencias Agropecuarias, Universidad Autónoma del Estado de Hidalgo, Tulancingo de Bravo, Hidalgo, México
[2] Unidad de Investigación Médica en Inmunología, Unidad Medica de Alta Especialidad, Hospital de Pediatría, Centro Médico Nacional "Siglo XXI", Instituto Mexicano del Seguro Social, Ciudad de México, México
[3] Área Académica de Medicina, Instituto de Ciencias de la Salud, Universidad Autónoma del Estado de Hidalgo, Pachuca, Hidalgo, Mexico
[4] Área Académica de Ingeniería en Alimentos e Ingeniería Agroindustrial. Instituto de Ciencias Agropecuarias, Universidad Autónoma del Estado de Hidalgo, Tulancingo, Hidalgo, México
* These authors contributed equally to this work.

Corresponding authors
Monica Salas-Rojas,
mony_salas@yahoo.com.mx
Deyanira Ojeda-Ramírez,
dojeda@uaeh.edu.mx

## ABSTRACT

Sea urchins are a group of benthic invertebrates characterized by having rigid globose bodies, covered in spines, and have an innate immune system that has allowed them to survive in the environment and defend against many pathogens that affect them. They are consumed for their unique flavor, but also for possessing a rich source of bioactive compounds which make them a source for a wide array of medicinal properties. Thus, these may be used to discover and develop new drugs such as anti-bacterials, anti-carcinogenics and anti-virals. Precisely for those reasons, this revision is centered on the known biological activities in various sea urchin species. Recently, the potential pharmacological benefits of nine sea urchin species [*Diadema antillarum* (Philippi 1845), *Echinometra mathaei* (de Blainville), *Evechinus chloroticus* (Valenciennes), *Mesocentrotus nudus* (Agassiz, 1863), *Paracentrotus lividus* (Lamarck, 1816), *Scaphechinus mirabilis* (Agazzis, 1863), *Stomopneustes variolaris* (Lamarck, 1816), *Tripneustes depressus* (Agassiz, 1863), and *Tripneustes ventricosus* (Lamarck, 1816)] have been evaluated. Our work includes a comprehensive review of the anti-fungal, anti-parasitic, anti-inflammatory, hepatoprotective, anti-viral, anti-diabetic, anti-lipidemic, gastro-protective and anti-cardiotoxic effects. Furthermore, we revised the compounds responsible of these pharmacological effects. This work was intended for a broad readership in the fields of pharmacology, drugs and devices, marine biology and aquaculture, fisheries and fish science. Our results suggest that organic extracts, as well as pure compounds obtained from several parts of sea urchin bodies are effective *in vitro* and *in vivo* pharmacological models. As such, these properties manifest the potential use of sea urchins to develop emergent active ingredients.

## INTRODUCTION

Sea urchins are marine invertebrates belonging to the phylum Echinodermata, subphylum Echinozoa and the class *Echinoidea*, which includes over 700 species worldwide (*Reich et al., 2015*; *Luparello et al., 2020*). Echinoderms belong to the group of deuterostomes, which relates them closely to vertebrates, and share some families of immunological genes with latter (*Li et al., 2015*; *Coates et al., 2018*). The class *Echinoidea* is characterized by having a body globose to flattened and discoid, stereom plates joined by connective tissue and calcite to form a rigid test, various pedicellariae types, mobile spines mounted on tubercles, and radial water vessels completely closed within the test (*Brusca, Moore & Shuster, 2016*). The coelomic fluid is isosmotic with seawater; the major role of this fluid is for circulatory purposes and containing different kinds of coelomocytes with a variety of functions, with one of them being an immunological role. The coelomic fluid contain factors (trophic and activating) that produce the coelomocytes themselves (*Brusca, Moore & Shuster, 2016*; *Pinsino & Matranga, 2014*). In addition, the mouth is surrounded by the peristomal membrane, and the anus is located on the opposite side of the mouth in a region known as the periproct with the ocular and genital plates (*Gomes et al., 2014*).

Moreover, marine ecosystem host the largest population of microbes and viruses in our planet. For example, the Indian Ocean has been found to have a bacterial abundance of $6 \times 10^4$ to $2.5 \times 10^6$ cells/mL in surrounding seawater (*Helber et al., 2018*). In addition, it is estimated that there are $10^{30}$ viruses in the oceans with an average count of $10^7$ viruses/mL of surface seawater (*Weynberg, 2018*). Therefore, benthic organisms, like sea urchins, are frequently exposed to these type of pathogens (*Franco, Patiño & Ortiz, 2015*; *Chiaramonte et al., 2019*; *AbouElmaaty et al., 2020*). Furthermore, they are also exposed to environmental pressures including hypothermia, hypoxia and changes in pH, xenobiotics, nanoparticles and various chemical substances, as well as UV radiation and free radicals (*Pinsino & Matranga, 2014*; *Chiarelli et al., 2016*). Despite not having an adaptative immunity and living in different conditions, sea urchins have survived until today with the help of their developed immune system (*Chiaramonte et al., 2019*). It should be noted that this immunity is provided by proteins with hemolytic and haemagglutinating properties (*Soboleva et al., 2018*), which have allowed this group of marine animals to acquire different genetic diversification abilities in order to adjust to the constantly changing environmental factors they face (*Oren et al., 2019*). Further, this immune system is enhanced by operating in conjunction with other strategic cellular and molecular mechanisms, such as protein synthesis against thermic shock and metallothioneins (which contain detoxicating and antioxidant properties, respectively), and in the event of extensive or persistent damage, they have the capacity for apoptosis and autophagy if a regeneration process is needed (*Chiarelli et al., 2016*). These collective abilities have allowed these organisms to develop complex molecular elements to use in diverse biological processes; hence, as benthic organisms, they are also considered a rich source of bioactive compounds with the potential for the discovery and development of new pharmacological compounds (*Pastrana-Franco, Santafé-Patiño & Quirós-Rodríguez, 2016*; *Luparello et al., 2020*; *Strahsburger et al., 2020*).

Sea urchins have long been used in Asia-Pacific regions as an alternative method to treat some diseases such as cardiovascular conditions and cancer (*Rahman, Arshad & Yusoff, 2015*). There has been an increase in their collection and consumption, due to their antibiotic, antiviral, antiprotozoal and antifungal agents (*Luparello et al., 2020*). It should be noted that the composition of these valuable elements varies considerably among the different species of sea urchins, not only because of their innate immunity, but due to their natural diet and physiological processes (*Stabili et al., 2018*). Recently, *Sibiya et al. (2021)* published a bibliographic revision focused solely on the anticarcinogenic, antioxidant, anticoagulating and antibacterial properties exhibited by sea urchins; however, these organisms exhibit others important pharmacological properties which were disregarded. Thus, the objective of this article was to create a systematic literary revision identifying the pharmacological properties found within sea urchin extracts and/or compounds recently reported that were omitted from the previous publication, which could be a helpful precursor to systematic reviews of pharmacological properties of sea urchins. This work was intended for a broad readership in the fields of pharmacology, drugs and devices, marine biology and aquaculture, fisheries and fish science.

## SURVEY METHODOLOGY

An organized search of sea urchins was carried out in terms of pharmacological activities attributed to their compounds, as well as preclinical studies performed. Data were collected as previously describe in *Vázquez-Atanacio et al. (2021)*. Specifically, related MeSH (Medical Subject Headings) terms were defined: "Pharmacological action", "Bioactivity compounds", "Hepatoprotector", "Antidiabetic", "Antifungal", "Antiviral", "Antilipidemic", "Anti-inflammatory" and "Antiparasitic", then each term was combined with "Sea Urchin".

Inclusion criteria were all articles found in the scientific information sources ScienceDirect, Scopus, Pubmed and Springer link were considered. A selection of titles was made, from which the abstracts were read and those that met the necessary characteristics were retrieved. The following criteria were included for the selection of documents: (1) In the case of pharmacological reports, documents exposing the use of different parts of sea urchins were selected. (2) Regarding *in vitro* and *in vivo* studies, articles were selected that mentioned in their methodology the type of test used, the species studied, and the type of extract or extracts used, as well as the compounds evaluated and their possible mechanism of action. Articles that did not meet the requirements were discarded.

### Pharmacological properties of sea urchins

As of the present investigation, we found reports about the pharmacological potential of nine sea urchin species according to inclusion criteria. Results indicated biological effects from organic and aqueous extracts obtained from sea urchin gonads, spines, intestine and test, as well as their entire bodies (Fig. 1); which produced bioactive peptides, terpenoids, phenolic acids, PUFAs and naphthoquinones. These, in turn, have exhibited antifungal, anti-parasitic, anti-inflammatory, hepatoprotective, antiviral, antidiabetic,

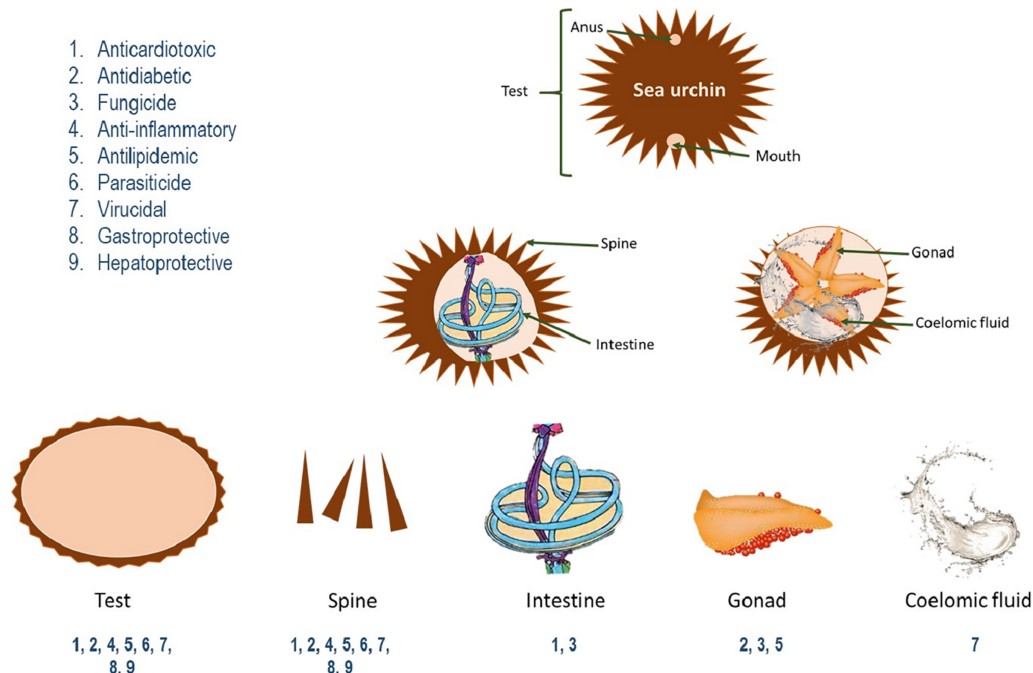

**Figure 1 Pharmacological activities of parts of sea urchins.**

antilipidemic, gastroprotective and anti-cardiotoxic effects in *in vitro* and *in vivo* models (Table 1). The following discussion describes, in alphabetical order, the pharmacological properties reported in these studies.

## Anti-cardiotoxic activity

The term cardiotoxicity refers to conditions of the heart such as arrhythmia, acute changes in contractile function, morphological and/or structural damage of the myocardia primarily because of the use of certain anticarcinogenic medication such as anthracyclines, Herceptin (©Roche, Basel, Switzerland), generically known as Trastuzumab, vincristine or doxorubicin, to name a few (*Wilkinson, Sidaway & Cross, 2016*). The manifestation of these side effects highlights the necessity for finding compounds that mitigate these harmful effects. A viable option is echinochrome A (6-ethyl-2, 3, 5, 7, 8,-pentahydroxyl-1, 4-napthoquinone, Ech A) (Fig. 2A) a naphtoquinoid pigment isolated from *Scaphechinus mirabilis* (Agazzis, 1863), which, at a concentration of 0.1–3 μM, significantly prevents cell death induced by cardiotoxic agents found in Tert-butyl hydroperoxide, disodium nitroferricyanide and doxorubicin in cardiac myoblast cells H9c2 in mice (*Jeong et al., 2014*). Depolarization of mitochondria, loss of the electrochemical gradient (ΔΨ), and increased mitochondrial reactive oxygen species (ROS) formation are universal processes associated with cell death (*Ježek, Cooper & Strich, 2018*; *Matsuyama & Reed, 2000*). Ech A inhibits increases in ROS and recovers depolarization potential of the mitochondrial membrane, produced by cardiotoxic agents in cardiac myoblast cells, thereby impeding, in a similar fashion, a decrease in cellular ATP (*Jeong et al., 2014*).

**Table 1 Biological activities of sea urchins.**

| Specie | Extract/compound | Biological activity | Assay | Important results | References |
|---|---|---|---|---|---|
| *Diadema antillarum* | Whole body dichloromethane extract | Antifungal | Disc diffusion | Inhibition zones of 13 to 18 mm against *Fusarium* sp. | *Franco, Patiño & Ortiz (2015)* |
| *Echinometra mathaei* | Extracts obtained with PBS from the spine and test | Antiparasitic | Protoscolices viability test. | 93.33% protococci killed | *Navvabi et al. (2019)* |
| *Evechinus chloroticus* | Crude extract and fractionated naphtoquinones pigments from spines and test | Anti-inflammatory | Cotton pellet-induced granuloma in adult male Sprague-Dawley rats | 12% to 52% inhibition of mouse granuloma. | *Hou et al. (2020)* |
| *Mesocentrotus nudus* | Lyophilized gonads | Antilipidemic | Mice with hyperlipidemic diet | Stabilization of aspartate aminotransferase (AST) and alanine aminotransferase (ALT) levels in serum, as well as liver weight and visceral fat reduction. | *Yamamoto et al. (2018)* |
| *Paracentrotus lividus* | Methanolic extracts obtained from the soft tissues (gonad, mouth and intestine) Echinochrome A isolated from test and spines | Antidiabetic | Streptozotocin induced Diabetes mellitus type 1 and 2 in rats | Gonads and intestine extracts were capable to reduce blood glucose concentrations in rats with DMT1 and DMT2 by values ranging between 138.67 and 180.33 mg/dL, respectively | *Soliman (2016), Soliman, Mohamed & Marie (2016)* |
| | Echinochrome A isolated from test and spines | Antilipidemic | Male rats with hyper-caloric diet | Levels of lipids, AST and ALT were normalized. | *Fahmy et al. (2019)* |
| | | Hepatoprotective | Acute sepsis lesions induced by cecal ligation and puncture method in male rats | Decrease of serum concentration of liver damage markers | *Mohamed et al. (2019)* |
| | Echinochromes | Gastroprotective | Induction of severe gastric ulcers in rats | Decrease of gastric juice volume (45%) and acidity of gastric juice. | *Sayed, Soliman & Fahmy (2018)* |
| *Scaphechinus mirabilis* | Echinochrome A | Anti-cardiotoxic | Quantitative fluorescence assay with CellTox Green cytotoxic assay | Prevents cardiotoxic-induced cell death in H9c2 cardiac myoblast cells in mice | *Jeong et al. (2014)* |
| | Aqueous extracts from viscera and spines | Antidiabetic | *In vitro* α-amylase enzyme inhibition | 19.41–42.46% of inhibition | *Chamika et al. (2021)* |
| *Stomopneustes variolaris* | Diterpenoid isolated from the gonads | Anti-inflammatory | *In vitro* COX and LOP enxymes inhibition | IC$_{50}$ = 2.37 and 2.01 mM for COX-2 and LOP-5, respectively | *Francis & Chakraborty (2020)* |
| *Tripneustes depressus* | Thermostable fractions of coelomic fluid | Antiviral | Viral inhibition test *in vitro* | 74% inhibition for Suid herpesvirus type 1 and 99% inhibition of rabies virus activity at 0.7 IU/mL | *Salas-Rojas et al. (2014)* |
| *Tripneustes ventricosus* | Ethanolic extract of the mix of soft tissus | Antifungal | Broth microdilution | 50% inhibition against *C. albicans* and *C. parapsilosis* at 8.6–17.4 μg/mL | *Velar Martínez et al. (2016)* |

In addition, Ech A decreases mitogen-activated protein kinases (MAPKs) pathway activation induced by cardiotoxic agents. MAPKs signaling pathways regulate a variety of biological process, including apoptosis, through multiple cellular mechanisms (*Jeong et al.,*

**Figure 2 Chemical structures of compounds isolated from several sea urchins.** (A) 6-ethyl-2,3,5,7,8-pentahydroxy-1,4-naphthoquinone (Echinochrome A). (B) Gallic acid. (C) Chlorogenic acid. (D) 4-hydroxyl-1-(16-methoxiprop-16-en-15-yl)-8-methyl-21, 22-dioxatricyclo [11.3.1.1$^{5,8}$] octadecane-3,19-dione. (E) 2,3,5,7-Tetrahydroxy-1,4-naphthoquinone (Spinochrome B). (F) 2,3,5,6,7,8-Hexahydroxy-1,4-naphthalenedione (Spinochrome E). (G) 3-Acetyl-2,5,7,8-tetrahydroxy-1,4-naphthoquinone (Spinochrome A).

*2014*; *Yue & López, 2020*). MAPK can regulate pro- and anti-apoptotic mechanisms depending on the stimulus, and its downstream targets include several transcription factors whose activities are enhanced by phosphorylation (*Yuan et al., 2013*; *Jeong et al., 2014*; *Yue & López, 2020*). Ech A was able to prevent phosphorylation of extracellular-regulated kinase (ERK1/2), C-Jun N-terminal Kinase (JNK), and p38 MAPK, which play a pivotal role in cell death. JNK helps with p38 regulating transcription factors, decreasing expression of anti-apoptotic proteins, and increasing expression of pro-apoptotic proteins, in addition to regulating the intrinsic and extrinsic apoptotic pathways (*Yuan et al., 2013*). On the other hand, it is known that activation of ERK1/2 is associated with anti-apoptotic functions through the control of cell proliferation and differentiation. However, there are several reports showing that this signaling pathway can also led apoptosis (*Yue & López, 2020*). In this regard, ERK1/2 activate caspase and pro-apoptotic protein in Bcl-2 family and deactivated the Akt signaling pathway can lead to apoptosis induction (*Tangchirakhaphan et al., 2018*).

Although only one study reported this activity with sea urchins, the results obtained by researchers demonstrated the therapeutic potential of these pigments to minimize adverse

cardiotoxic effects produced by some of the clinical drugs presently in use. As such, these may be used in the future to develop new drugs to prevent cardiac risks.

## Antidiabetic activity

Diabetes mellitus is a global disease which has increased rapidly in recent years. In 2015 there were an estimated 415 million persons afflicted. Furthermore, it is estimated that by the year 2040 that number will increase to 642 million individuals. It is the principal cause for morbidity and mortality in the world. Diabetes mellitus Type I (DMT1) as well as Type 2 (DMT2), cause a wide array of complications such as blindness, chronic renal failure (nephropathy), neuropathy, amputations not resulting from trauma, increased risk for cardiac arrest and ischemic stroke (*Rojas-Martínez et al., 2018*; *Zheng, Ley & Hu, 2018*). Thus, this pathology should be managed adequately or otherwise may gravely affect the quality of life of patients. In this context, studies reveal that Ech A (Fig. 2A) derived from the spines and test of sea urchins, as well as methanol extracts obtained from a mix of soft tissues, namely the gonads and intestine of *Paracentrotus lividus* (Lamarck, 1816), at a dose of 1 and 500 mg/kg, respectively, are capable of significantly reducing blood glucose concentrations in rats with DMT1 and DMT2 by values ranging between 138.67 and 180.33 mg/dL, respectively. In addition, methanol extracts increased blood insulin concentration almost twofold. The authors attribute the ameliorative effect of the methanolic extract to the presence of saponins, which have similar properties as insulin, stimulating glucose absorption uptake enhancing Glut4 expression and thereby contributing to its storage in adipocytes (*Soliman, 2016*; *Soliman, Mohamed & Marie, 2016*). While the hypoglycemic effect of Ech A is due to pancreas regeneration, insulin production increase, improvement in glucose homeostasis and decrease in insulin resistance (*Mohamed, Soliman & Marie, 2018*).

Along those same concepts, the enzyme α-amylase is key in managing diabetes since it is involved in carbohydrate decomposition within the human body. Inhibiting its availability within the body would correlate with increased postprandial blood glucose levels after carbohydrates consumption. In this respect, extracts obtained from the spines, test, gonads and intestines of *Stomopneustes variolaris* (Lamarck, 1816) can inhibit α-amylase (19.41–42.46%); of these extracts, those from the intestine obtained at a temperature of 110 °C and at 150 °C from the spines, are the most effective. An HPLC analysis of active extracts show gallic acid (Fig. 2B) and chlorogenic acid (Fig. 2C) as majority components (*Chamika et al., 2021*). Phenolic acids are capable of inhibiting α-amylase through non-covalent interactions. In particular, the importance of hydroxyl groups in polyphenols have been reported in the interaction with amino acid residues at the active site of α-amylase (Glu233) *via* hydrogen binding. In addition, the presence of C=C double bond conjugated with a carbonyl group in chlorogenic acid structure stabilizes the binding forces to the active site of the α-amylase (*Aleixandre et al., 2022*).

On the other hand, diabetic nephropathy (DN) is one of the main complications in diabetic patients. DN is characterized by hemodynamic alterations in renal microcirculation and structural changes in glomerulus as evidenced by the significant elevation in urea level related to chronic hyperglycemia (*Mohamed, Soliman & Marie,*

*2018*). Several researchers have reported an overproduction of ROS as the common denominator linking the altered metabolic pathways in the kidneys with disrupted renal hemodynamics. Renal stress is a result of an upregulation between antioxidant defense (superoxide dismutase (SOD): manganese SOD and copper/zinc SOD; glutathione system: glutathione peroxidase and glutathione reductase; catalase (CAT); and coenzyme Q) and ROS production mechanisms (xanthine oxidase, cytochrome P450 systems, uncoupled NO synthase (NOS), mitochondrial respiratory chain, and NDPH oxidases (NOXs)) in the kidney (*Jha et al., 2016*). In addition, renal NOS expression and activity are increased at early diabetes; however, they are decreased in prolonged diabetes leading to vascular NO deficiency, which may induce the progression of DN (*Soliman, Mohamed & Marie, 2016*). Ech A was able to decrease kidney damages caused by diabetes in rats. DMT1 and DMT2 rats treated with Ech A (1 mg/kg) showed decreased serum creatinine, urea, and uric acid, as well as regeneration in glomerulus and Boyan's capsule with normal renal tissue architecture. In addition, an increase in SOD, GST, and CAT activities and NO and GSH concentration, as well as decrease malondialdehyde (MDA) concentration was observed. These results evidenced that Ech A improves renal function in diabetic rats *via* regulation of renal oxidative stress and concomitant improvement of glucose metabolism due to insulin activates NOS by protein kinase B (PKB)-mediated phosphorylation in endothelial cells increasing NO concentration (*Soliman, Mohamed & Marie, 2016*).

## Antifungal activity

Some fungi affect plants and animals, including humans. One of the primary pathogenic fungi of humans is *Candida albicans*, this saprobic fungi is able to colonize mucous membranes resulting in vaginal, respiratory and intestinal infections (*Shao et al., 2019*). Extracts from native marine invertebrates have shown antifungal properties. Four studies on sea urchins report these findings. In this case, ethanolic extracts from a mix of soft tissue of *Tripneustes ventricosus* (Lamarck, 1816) showed antifungal effects against *C. albicans* ATCC 10231 using $8.6 \pm 2.8$ µg/mL and $17.4 \pm 0.6$ µg/mL (LD$_{50}$) for *Candida parapsilosis*. It is worth noting that the extract effects towards *C. albicans* was similar to the drug Fluconazole (IC$_{50}$ $8.0 \pm 0$ µg/mL) (*Velar Martínez et al., 2016*).

On the other hand, phytopathogenic fungi cause great damage to crop resulting in significant economic losses within the food industry. Nowadays, synthetic fungicides are used to prevent widespread contamination of food sources, but these fungicides also contaminate food products and are harmful to human health (*Hernández-Montiel et al., 2018*). A methanol, dichloromethane and aqueous extract obtained from *Diadema antillarum* (Philippi 1845) were assessed against strains of phytopathogenic fungi *Sclerotium sp.*, *Rhizoctonia sp.* and *Fusarium* sp.; dichloromethane extract was the only extract effective against *Fusarium* sp. (IZ = 13 and 18 mm at 100 and 2,000 µg/mL, respectively) (*Franco, Patiño & Ortiz, 2015*).

Saponins compounds have a saccharide moiety connected glycosidically to sapogenin (hydrophobic aglycone), which has a triterpene or steroid backbone. These biologically active compounds occur in terrestrial plants and some marine invertebrates, particularly echinoderms, octocoral and sponges (*Bahrami & Franco, 2016*). There are several reports

about fungicidal mechanisms of saponins. For instance, saponins can interact with fungal plasma membrane, disrupting its integrity, and increasing the permeability to hydrophilic compounds and, as result, can cause ions leakage, changes in cell membrane potential, and intracellular osmolarity which may lead to death of fungi. In addition, steroidal saponins can form complexes with sterols in fungal membranes and cause loss of fungal membrane integrity. Additionally, it has been reported that saponins inhibited yeast-to-hyphal transition and prevent biofilm formation in *C. albicans*, which is one of the most important virulence factors employed by the fungi to establish infection in the host (*Budzyńska et al., 2014*; *Barros-Cota et al., 2021*).

## Anti-inflammatory activity

Inflammation is a specific and controlled defense mechanism triggered by the immune system in response to pathogens, lesions or tissue damage. Generally, an inflammatory response is tasked to address and resolve infections or repair tissue damage and wounds. An unregulated inflammatory response disrupts cellular homeostasis leading thereby to the production of increased amounts of ROS, reactive nitrogen species (RNS) and various proteases which may induce tissue damage, fibrosis, cellular proliferation, chronic inflammation, inflammatory trauma and chronic systemic damage (*Maleki, Crespo & Cabanillas, 2019*).

Ciclooxigenases (COX) and lipooxigenases (LOP) are two enzymes families that are key in the inflammatory process. Both produce lipid derivatives (prostanoids and leukotrienes) from arachidonic acid that act as chemical mediators at the site of inflammation. There are two distinct COX isoforms, COX-1 and COX-2. The first one is constitutively expressed in most cells and is the dominant source of prostanoids that subserve housekeeping functions; while COX-2 is induced by inflammatory stimuli, hormones and growth factors, is generally assumed to be more important source of prostanoid formation inflammation (*Wang et al., 2021*). Reports indicate that 4-hydroxyl-1-(16-methoxiprop-16-en-15-yl)-8-methyl-21, 22-dioxatricyclo [11.3.1.15,8] octadecane-3,19-dione (Fig. 2F), a diterpenoid isolated from the gonads of *S. variolaris*, inhibits COX-2 ($IC_{50}$ = 2.37 mM) and 5-lipoxygenase (LOP-5) ($IC_{50}$ = 2.01 mM) enzymes. Further, this compound possesses a selectivity index of 1.25 for COX1/COX2, which shows its specificity for the pro-inflammatory enzyme, and molecular models reveal the terpenoid attaches to the active site of the enzyme forming hydrogen bonds with Leu131.A and Arg319.B residues (*Francis & Chakraborty, 2020*).

Furthermore, predict bioavailability properties of bioactive molecules is crucial for the drug design and there are physicochemical parameters used to do so. For example, topological polar surface area (TPSA), which is the sum of the contributions to the molecular (usually van der Waals) surface area of polar atoms and their attached hydrogens; high values of TPSA indicate high rate of solubility of solids and favored certain cellular transports (*Prasanna & Doerksen, 2009*). Another parameter is the logarithm of octanol–water partition coefficient (log POW), which is used to express the lipophilicity of a compound, values lesser than five are preferred for a compound for sustaining a steady hydrophobic–lipophilic equilibrium (*Amézqueta et al., 2020*). Anti-inflammatory

membrane-type diterpenoid 2f isolated from *S. variolaris* showed a lesser hydrophobic character (log POW = 0.34) that α-tocopherol (log POW = 9.98) and tPSA value (82.06) greater than ibuprofen and tocopherol (37.3 and 29.46, respectively), which propose that this compound could be more disponible for its distribution and could have high cell membrane permeability (*Francis & Chakraborty, 2020*).

On the other hand, ethyl acetate extracts obtained from the spines and test of the sea urchin *Evechinus chloroticus* (Valenciennes) that contain mainly EchA (Fig. 2A) and spinochromes E and B (Figs. 2G and 2F, respeively), demonstrate anti-inflammatory response towards mice granulomas (I% = 52.01 at 8 mg). Ech A decrease NFκB and TNF expression and reduced IL-6, 6-keto-PGF1α, PGE2, and NO production (*Hou et al., 2020*), while, spinochromes E and B can interact with the histamine receptor H1R binding to the receptor active site through its hydroxyl groups and, therefore, inhibit its recognition by histamine, blocking the signaling pathway and avoiding the formation of some chemical inflammation mediators as prostacyclins, NO, arachidonic acid (and its metabolites), and thromboxanes, as well as the increase of chemotaxis of eosinophils and neutrophils at the site of inflammation (*Pozharitskaya et al., 2013*; *Branco et al., 2018*).

## Antilipidemic activity

Dyslipidemia is having a high concentration of lipids in an organism or the blood, caused by enzymatic congenital defects (primary dyslipidemia), or may occur as a result of diseases such as diabetes mellitus, hypothyroidism, chronic kidney conditions, bile obstruction, pancreatitis, gout, alcoholism and various hepatic disorders (secondary dyslipidemia). In both cases this disease results in significant risks for individuals. The drugs presently used in the treatment of dyslipidemia are effective, but may have adverse secondary effects (*Yang et al., 2018*). Therefore, there is a need to seek other forms of treatment with less harmful secondary effects. For example, the lyophilized gonads of *Mesocentrotus nudus* (A. Agassiz, 1863) (500 mg/kg body mass) administered to mice which were previously provided with high fat content diets, showed a significant reduction in their overall body weight (from 35.97 to 29.8 g). Liver weight also decreased from 1.12 to 0.88 g and visceral fat from 3.38 to 2.0 g. Further, the administration of lyophilized gonads to mice, stabilized the enzymatic levels of aspartate aminotransferase (AST) and aminotransferase alanine (ALT) to 33 and 13.8 U/L, respectively, and attenuated triacylglycerol hepatic accumulation and increase levels of polyunsaturated fatty acids (PUFAs), arachidonic and eicosatetraenoic (EPA) acids in the liver. In addition, sea urchin gonads induced the mitochondrial uncoupling protein (UPC-1) expression in brown adipose tissue, whose specialized function is to generate thermogenic heat, a preventative effect against lipid accumulation in the body and liver. The authors suggest that these pharmacological effects are due the presence of PUFAs in sea urchin gonads. PUFAs, especially EPA and docosaesaenoic (DHA) acids (both omega-3 fatty acids), regulate the expression of lipid and energy metabolism-associated genes. For instance, they are able to downregulate the expression of hepatic CD36, a fatty acid transporter, which is enhanced in rodents with fatty livers (*Yamamoto et al., 2018*). Furthermore, PUFAs increase the activity of UPC-1. This mitochondrial protein mediates the FA-dependent $H^+$

influx resulting in the uncoupling of electron transport from ATP synthesis that leads to the rapid production of heat and have a higher conductance in presence of PUFA: linoleic, retinoic, and arachidonic acids (*Beck et al., 2007*).

Similar results were observed in rodents that were fed specific echinochromes derived from the test and spines of *P. lividus* (1 mg/kg of body weight). Rodents had been provided a hyper-caloric diet, after consuming echinochromes, normal levels of lipids (272 to 92 mg/dL), cholesterol (from 184.67 to 121.45 mg/dL), LDL (from 166.62 to 98.62 mg/dL), ALT (from 54.17 to 37.50 UI/mL) and AST (from 197.5 to 113.67 UI/mL) were restored. The hypolipidemic effect of echinochromes may be through its inhibition of cholesterol and triglyceride synthesis key enzymes (*Fahmy et al., 2019*).

## Antiparasitic activity

Cystic echinococcosis is a zoonosis caused by cestode larvae of the genus *Echinococcus*, these parasites affect canids, a wide range of domestic ungulates and even humans. In livestock farming worldwide, annual economic losses are estimated at billions of US dollars due to this parasitosis (*Ohiolei et al., 2019*). Extracts obtained with PBS from the test and spines of *Echinometra mathaei* (de Blainville) have shown scolycidal effects against hydatid cysts produced by *Echinococcus* spp. Both extracts showed maximum effects at a concentration of 20 µg/mL, killing 78.01% and 93.33% of protoscolices, respectively, after 60 min of exposure. In addition, the extract from spines showed 80% activation of caspase-3, while the extract from tests showed 60% activation. Caspase-3 is a key effector in the process of apoptosis, acting by amplifying the caspase activator signal (such as caspase-8) (*Navvabi et al., 2019*). This executioner enzyme breaks down proteins including poly(ADP-ribose)polymerase family member-16 (PARP-16); procaspases 6, 7 and 9; protein kinase C gamma; and delta and beta catenin, leading to cell death. Furthermore, caspase-3 is indispensable for apoptotic chromatin condensation and deoxyribonucleic acid fragmentation in all cells and is required for dismantling the cell and development of apoptotic bodies (*Yadav et al., 2021*). The authors suggest scolicidal effects of *E. mathaei* may be due to presence of polyhydroxylated naphthoquinone pigments such as Ech A in the extracts; however, this is not clear yet.

## Antiviral activity

Various pathogens such as viruses cause many diseases for humans and can even generate pandemics such as the recent outbreak we are experiencing today by coronavirus (SARS-CoV-2), which has already taken over the world causing severe respiratory diseases and even the death of thousands of people. Due to viruses constantly mutating and generating resistance to drugs, the search for new effective treatments against them is a great challenge (*Olascoaga-Del Angel et al., 2018*; *Baig et al., 2020*). In this context, there is an urgent need to find specific antiviral drugs from natural molecules. In this regard, the pigments Ech A (Fig. 2A) and Spinochrome A (Fig. 2G) have been evaluated *in silico* for their ability to inhibit the spike (S) protein of the SARS-CoV-2 virus. This protein is essential to mediate the fusion of the virus with the cell membrane and make possible its entry into the cell, so it has become a main target for the development of drugs against this

virus. These compounds have shown promising results, since they can interact with virus proteins through non-covalent interactions. Ech A showed 3 hydrogen bonds and 32 Van der Walls contacts with different residues of the protein, while the spinochrome had 5 hydrogen bonds and 32 Van der Walls contacts, all without collisions (*Barbieri et al., 2020*).

Furthermore, a study carried out on the sea urchin *Tripneustes depressus* (Agassiz, 1863), revealed that the thermostable fractions at 56 °C and 72 °C of the peptides present in the coelomic fluid possess antiviral activity against Suid herpervirus type 1 (SHV-1) and rabies virus (RV). Reaching 74% inhibition for the first virus and 99% inhibition of RV viral activity. In addition, highest inhibition against both viruses was observed when coelomic fluid was incubated with them prior to addition of cells, which reveladed that coelomic fluid worked directly with the virus and de-stimulated the cell to produce molecules with antiviral effect. The authors suggest that peptides present in coelomic fluid may act by interfering with the virus-cell junction, either *via* membrane destabilization or receptor blockage (*Salas-Rojas et al., 2014*).

## Gastroprotective activity

Gastritis and peptic ulcer are very frequent conditions in the human population, with a general prevalence of 5% to 10% and an incidence between 0.1% and 0.3% per year. These pathologies are generated by a hypersecretory environment, dietary factors (poor nutrition), stress, *Helicobacter pylori* infection or widespread use of non-steroidal anti-inflammatory drugs (NSAIDs), which can cause abdominal pain, nausea, vomiting, weight loss and bleeding or perforation and/or mucosal rupture in the stomach, proximal duodenum, esophagus and even Meckel's diverticulum (*Lanas & Chan, 2017*; *Narayanan, Reddy & Marsicano, 2018*).

Ech A (Fig. 2A), isolated from *P. lividus*, has shown a favorable effect on rats with indomethacin-induced gastric ulcer and cold stress, decreasing gastric juice volume from 3.48 to 2.2 and 1.94 mL at doses of 5 and 10 mg/kg and gastric juice acidity (from 141. 6 to 28.4 and 10.8 mEq/L), as well as the percentages of presence and size of peptic ulcer lesions between 73.51% and 75.50%. In addition, it also significantly adjusted the levels of reduced glutathione (GSH) and glutathione-S-transferase (GST) biosynthesis enzymes (60.28–62.14 mg/g for GSH and 0.57–0.65 U/g for GST). The anti-ulcerogenic effect of Ech may be due to its antioxidant property, stimulating cell growth and repair of the gastric mucosa, and it is also believed that it may minimize damage by inhibiting or regulating the production of proinflammatory cytokines (*Sayed, Soliman & Fahmy, 2018*).

## Hepatoprotective activity

Liver failure is a frequent and often fatal medical emergency. The incidence of this disease is rising due to the increase in alcohol and drug consumption, the growing epidemic of obesity and diabetes mellitus, as well as complications from sepsis (*Sarin & Choudhury, 2016*). The latter is a progressive disorder of multiple organ dysfunction caused by severe bacterial infection, especially of the abdominal cavity. During this process, the liver generates an overproduction of proinflammatory cytokines in response to the infection

(*Mohamed et al., 2019*). Therefore, molecules with anti-inflammatory and antioxidant capacity are potent hepatoprotective candidates.

Ech obtained from test and spines of *P. lividus*, has been shown to have a hepatoprotective effect on acute lesions caused by sepsis induced by the cecal ligation and puncture method (CLP) in rats. The administration of Ech at a dose of 1 mg/kg orally for 2 days generated a decrease (40–70%) in the serum concentration of the markers of liver damage: serum alanine transaminase (ALT), gamma-glutamyl transferase (GGT), aspartate transaminase (AST), alkaline phosphatase (ALP) and lactate dehydrogenase (LDH), being greater the effect on the latter. Similarly, this naphthoquinone-type pigment had an effect on oxidative stress markers, generating a slight increase (5%) in the concentration of GSH and a decrease in the concentration of hydrogen peroxide ($H_2O_2$) (15%), nitric oxide (NO) (15%), and malondialdehyde (MDA) (50%); in addition to increasing the effect of liver antioxidant enzymes [catalase (CAT), superoxide dismutase (SOD), glutathione peroxidase (GPx) and glutathione-s-transferase (GST), glutathione reductase (GR)] by 20% to 50%, with the greatest effect on GR. On the other hand, the hepatocytes of the Ech-treated rats did not show hemolyzed blood cells or degenerated hepatocytes, compared to the CLP-operated group. The hepatoprotective action of Ech is related to its ability to inhibit ROS and stimulate the activity of endogenous antioxidant enzymes (*Mohamed et al., 2019*).

## CONCLUSIONS

Sea urchin-derived molecules are potential candidates for the development of effective drugs for the treatment of chronic degenerative diseases such as metabolic syndrome, diabetes or arthritis, as well as for the treatment of emerging and re-emerging infectious diseases in humans and animals. Furthermore, the reports included in this review show that the bioactive compounds are chemically diverse and not restricted to one body part, but instead evidenced that the entire organism and even its waste products, can be used. Regarding active compounds, terpenes, saponins, peptides, PUFAs and phenolic compounds have been described as molecules responsible for some biological activities exerted by sea urchins. However, in some cases, chemical studies to unequivocally identify the structure of the active principles are still lacking, as well as toxicological, preclinical and clinical trials to demonstrate their therapeutic safety and pharmacological efficacy, which constitutes a major challenge for the international scientific community.

### Funding
Moreno-García received funding from the National Council of Science and Technology (CONACYT) for the scholarship number 765944, received as support to conduct this investigation. The funders had no role in study design, data collection and analysis, decision to publish, or preparation of the manuscript.

## Grant Disclosures

The following grant information was disclosed by the authors:
National Council of Science and Technology (CONACYT): 765944.

## Competing Interests

The authors declare that they have no competing interests.

## Author Contributions

- Dulce María Moreno-García conceived and designed the experiments, performed the experiments, analyzed the data, prepared figures and/or tables, authored or reviewed drafts of the article, and approved the final draft.
- Monica Salas-Rojas conceived and designed the experiments, analyzed the data, authored or reviewed drafts of the article, and approved the final draft.
- Eduardo Fernández-Martínez performed the experiments, authored or reviewed drafts of the article, and approved the final draft.
- Ma del Rocío López-Cuellar conceived and designed the experiments, authored or reviewed drafts of the article, and approved the final draft.
- Carolina G. Sosa-Gutierrez performed the experiments, prepared figures and/or tables, and approved the final draft.
- Armando Peláez-Acero performed the experiments, prepared figures and/or tables, and approved the final draft.
- Nallely Rivero-Perez analyzed the data, prepared figures and/or tables, and approved the final draft.
- Adrian Zaragoza-Bastida analyzed the data, prepared figures and/or tables, and approved the final draft.
- Deyanira Ojeda-Ramírez conceived and designed the experiments, authored or reviewed drafts of the article, and approved the final draft.

## Data Availability

All data are described in our literature review.

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
