# Peer review of "Sea urchins: an update on their pharmacological properties"

_PeerJ, doi:10.7717/peerj.13606_

## Round 0.1 · original submission · Major Revisions

Your manuscript has been evaluated by three peer reviewers, and the reviewer comments are appended below.

I am glad to tell you that the reviews are positive. However, they are also advising that your manuscript should be revised at some places before being accepted (e.g. general presentation of sea urchins’ anatomy and evolution and lack of a detailed description of the mechanisms of action of compounds).

Based on the referees' recommendations, I arrive at this decision: The manuscript does merit publication in PeerJ but it is not acceptable in its current form and needs a major revision based on the reviews and my general editorial comments below. I therefore invite you to resubmit a revised version.

Please carefully consider the comments of the reviewers and provide a point-by-point response which clearly defines the changes made. Note that reviewer 2# has also provided his/her advice in an annotated pdf file.

Thank you for your patience with the evaluation process and for choosing PeerJ.

I look forward to receiving your revised manuscript.

Editor's general remarks
I suggest adding the author(s) of the scientific name and the year it was published when a species' name is cited for the first time in the manuscript as recommended by the International Code of Zoological Nomenclature (ICZN).

Reviewer 1 ·

Basic reporting

I do not have any comment on the English. I would just suggest to use "an update on" instead of "an update of".
The literature overview is quite broad and updated, I consider it sufficient to cover the topic.
In the paper there are no tables and figures. I think that this is a lack of the paper. In the reviews, particularly, this kind of schemes are very useful to guide and summarize the findings.

Experimental design

The review is well organized in a systematic way.

Validity of the findings

The main failure of the paper is the lack of detailed description of the mechanisms of action of compounds showing a pharmacological / biological activity. The description is too general and lack in going deeper, but this could represent the most useful information for the reader. Most of the times the authors talk about "the extract effect" towards a specific activity, but frequently the cited papers go into more details of the specific components responsible of the activity.
More information should be added.
The conclusions should be expanded. In the present paper, they are just a general and too brief summary of the manuscript.

Reviewer 2 ·

Basic reporting

The manuscript #67602 by DM Moreno-García and co-authors is a synthesis of potential, medicinal properties of bioactive compounds found, in former works, in nine common species of sea urchins. Based on an extensive review of the literature, it highlights the potential of numerous, different compounds present in sea urchins with distinct potential pharmacological implications, from antifungal and anti-arasitic benefits, to anti-diabetic and cardiotoxic effects. I think this short, and very synthetic and rich review should be of interest to a wide panel of readers, from the fields of marine biology to pharmacology, and highlight the potential of these organisms to develop efficient active ingredients in the future.

The manuscript is clearly written in correct, professional English and well-structured. Only some terms refering to the anatomy of sea-urchins are not correctly used in the text and in figure 1. Please see my specific comments and suggestions on this point below.

The objectives of the review is well-presented in the introduction. However, there are significant weaknesses in the general presentation of sea urchins’ anatomy and evolution, along with imprecisions in the use of several anatomical terms and in the description of biological functions (see my comments below and in the annotated version of the ms).

Experimental design

The review method is rigorous and well described in a dedicated section (survey methodology). Sources are clearly cited and the review is logically organized following the potential parmacological effects.

Validity of the findings

While the check-list of active substances with potential pharmacological effects in sea urchins is rich, the conclusion very logically highlights that the understanding of some active principles is still needed without mentioning the need of toxicological and clinical trials before effective drugs and treatments can be developed.

Additional comments

To conclude, I find the manuscript topical and of high interest. My main concern is that the general presentation of sea urchins and their anatomy, and of their scientific value in the introduction comprises some mistakes and imprecisions. Please find below and in an annotated version of the manuscript (comments and corrections in pdf) my detailed remarks and suggestions on the form of the manuscript.

In the abstract :

L41 « primitive » : The word "primitive" should be avoided as it may refer to a concept of final evolution that has no place in modern evolutionary sciences. It could be replaced by "ancestral", for instance. However, I don't see how the immune system of echinoids should be regarded as more ancestral than the one of other groups such as asteroids, polychetes, or sponges...?

L42 « ever-changing environmental conditions » : it is not clear why sea urchins should be impacted by ever-changing conditions compared to other marine oganisms ?

In the introduction :

L59 : no phylogenetic analsyis ever supported an ancestral position of sea-urchins at the base of echinoderms. This is very surprising.

L60-63 : this statement is not correct. There are many invertebrates that have closer phylgenetic relationships with vertebrates than echinoderms (ex : all chordates). Please amend.

L64-65 : these are general characters present in many echinoderms, they are not specific to echinoids.

L 65- 75 : this section is confusing as some of the mentioned functions are associated to the water vascular system, others are in link with the hemal system, other (ex : locomotion) are independent. Can the authors be more specific and clearly highlight the biological and physiological roles of the hemal system in echinoids in link with the manuscript’s objectives.

L 78 : It is not clear why benthic organisms are more frequently exposed to pathogens compared to other organisms ?

L 83 : here again, the necessity of having evolved an efficient immunitory system is not specific to echinoids but common to all organisms

Figure 1.
In the figure, some terms related to the anatomy of sea urchins are not correct. Please replace the words « spike » for « spine » and « shells » for « test ». Spine is the official word to qualify the sea urchin’s appendices, and test the official term to qualify the internal skeleton of echinoderms (while shell refers to the external skeleton present in mollusks).

Annotated reviews are not available for download in order to protect the identity of reviewers who chose to remain anonymous.

Reviewer 3 ·

Basic reporting

The English quality is not the best and needs improvement.
The used references seem appropriate and thoroughly selected.
The article structure is adequate for a review.
I believe the review has a broad scope and has cross-disciplinary interest.
A similar review has been published in 2021 by Sibiya et al. but focused solely on the anticarcinogenic, antioxidant, anticoagulating and antibacterial properties exhibited by sea urchins derived molecules. This review is a complement focusing on antifungal, anti-parasitic, anti-inflammatory, hepatoprotective, antiviral, anti-diabetic, anti-lipidemic, gastro-protective and cardiotoxic effects.
The introduction seems appropriate but needs improvement according to the specific comments.

Experimental design

I believe this review is within the Scope and Aims of PeerJ.
The methodology used to select the papers to be included in this review are described.
The papers are adequately cited and the review is organized logically. However, some sentences are too long and recommendation are provided in the Detailed comments.

Validity of the findings

Conclusions are brief but seem adequate for a review.

Additional comments

General comments:

In this manuscript Moreno-García and colleagues provide a review on recent studies on sea urchin bioactive molecules and their potential pharmacological benefits such as antifungal, anti-parasitic, anti-inflammatory, hepatoprotective, antiviral, anti-diabetic, anti-lipidemic, gastro-protective and cardiotoxic effects. They stress that their main goal with this review is to complement a recent review from Sibiya and collaborators (2021) that focused solely on the anticarcinogenic, antioxidant, anticoagulating and antibacterial properties exhibited by sea urchins derived molecules.

The authors conclude that sea urchin derived molecules have a great potential for the development of effective drugs for many modern society human illnesses, but there is still a long way to go, since studies are lacking that elucidate the structure of the active principles, as well as toxicological, preclinical and clinical trials to demonstrate their therapeutic safety and pharmacological efficacy.

Overall, the manuscript content is relevant in view of the development of the Blue Economy. The English quality is not the best and needs improvement, has well as clarification in some minor points detailed below.

In my opinion, the current manuscript meets the quality guidelines to be published in PeerJ, after the proposed corrections (see specific comments).


Specific comments:

Line 59: replace by “…marine invertebrates of the class …”

Line 59: explain what do you mean by “the class Echinoidea at the base of the echinoderms”. The current phylogenetic tree of Echinodermata, support two sister groups – the Asterozoa (sea stars and brittle stars) and Echinozoa (sea urchins and sea cucumbers). See Reich et al. 2015. Phylogenomic Analyses of Echinodermata Support the Sister Groups of Asterozoa and Echinozoa.

Line 78: replace by “…exposed to pathogens…”

Line 83: replace by “…conditions sea urchins…”

Line 84: replace by “…innate immunitary system…”

Line 87: replace by “…animals to acquire…”

Line 91: you mention “apoptosis and autophagy” which are mainly related with echinoderm’s ability to regenerate. I think this should be clearly stated for the readers.

Line 98: replace by “…Asia-Pacific regions…”

Line 99: replace by “cardiovascular conditions…”

Line 103: replace by “…sea urchins…” and “but due to their…”

Line 120: replace by “Hepatoprotector”

Line 132: remove or add a Subheading

Lines 133-134: replace 09 by 9

Lines 138-139: in-vitro and in-vivo should be in italics

Line 152: explain the abbreviation ROS for non-specialized readers. This was done in line 213 but should be done here.

Line 158: replace “medicines” by drugs

Line 162: replace What’s more by Furthermore

Line 175: replace by “…improvement in glucose homeostasis and decrease in insulin resistance…”

Line 176-179: this should be a sentence on its own and should be better explained “Methanol extracts most likely contain saponins, which have properties similar to insulin, stimulating glucose absorption and thereby contributing to its storage in adipocytes (Soliman, 2016; Soliman, Mohamed & Marie, 2016).”

Line 185-186: replace by “…are the most effective.”

Line 191: replace by “…in vaginal, respiratory and intestinal infections”

Line 193: explain “…from the interior…”. From which part or organs were the extracts prepared?

Line 205: rewrite this sentence “which are be found in from other types of echinoderms”. Do you mean “which are known to be found…”?

Line 212-213: replace by “…leading thereby to the production of increased amounts of ROS, …”

Line 221: replace by “…attaches to the active site…”

Line 224: replace by “…that contain mainly …”

Line 229: replace by “…Dyslipidemia is having a high concentration of lipids in an organism or the blood…”

Line 237: replace by “…administered to mice which were previously provided with high fat content diets…”

Line 251: replace by “…preventive effect…”

Line 267-272: too long, divide in two sentences

Line 274: in silico must be in italic

Line 273-277: again too long, divide in two sentences

Line 279: replace “forces” by “contacts”

Line 325-327: replace by “…[catalase (CAT), superoxide dismutase (SOD), glutathione peroxidase (GPx) and glutathione-s-transferase (GST), glutathione reductase (GR)]”

Line 334: replace “Sea urchins” by “Sea urchin-derived molecules…”

Line 336-339: rephrase as follows “Furthermore, the reports included in this review, show that the bioactive compounds are not restricted to one body part, but instead evidence that the entire organism and even its waste products, can be used.”

Table 1: Scaphechinus mirabilis, Stomopneustes variolaris, Tripneustes depressus and T. ventricosus are not mentioned in the text only in Table 1. This should be corrected.

---

## Round 0.2 · accepted · Accept

I am pleased to inform you that your paper has been accepted for publication without further changes.

Thank you for submitting your work to PeerJ. We hope you consider us again for future submissions.

Reviewer 1 ·

Basic reporting

The authors met all the reviewers comments and clearly improved the menuscript. I consider it acceptable in the present form

Experimental design

The authors met all the reviewers comments and clearly improved the menuscript. I consider it acceptable in the present form

Validity of the findings

The authors met all the reviewers comments and clearly improved the menuscript. I consider it acceptable in the present form

Additional comments

The authors met all the reviewers comments and clearly improved the menuscript. I consider it acceptable in the present form

Reviewer 3 ·

Basic reporting

The authors have significantly improved the manuscript and have incorporated all the suggested revisions.

Experimental design

The authors have significantly improved the manuscript and have incorporated all the suggested revisions.

Validity of the findings

The authors have significantly improved the manuscript and have incorporated all the suggested revisions.